# Theorization regarding access to oral health care for Brazilian prisoners: A qualitative study

Naessa Santos Borges Zure[1], Glaubieny Lourenço dos Santos[2],
Thallys Rodrigues Félix [1], Jaqueline Vilela Bulgareli[3], Álex Moreira Herval [3]*

1 Post-Graduation Program in Dentistry, Universidade Federal de Uberlândia (UFU), Uberlândia, Minas Gerais, Brazil, 2 School of Dentistry, Universidade Federal de Uberlândia (UFU), Uberlândia, Minas Gerais, Brazil, 3 Division of Preventive and Community Dentistry, School of Dentistry, Universidade Federal de Uberlândia (UFU), Uberlândia, Minas Gerais, Brazil

* alexmherval@ufu.br

## Abstract

The Brazilian prison system has faced difficulties in guaranteeing the fundamental rights of the population deprived of liberty, including access to oral health. In view of this, the aim of our study was to formulate a theory concerning access to oral health and the influencing factors, grounded in the perspectives of incarcerated men. We carried out a qualitative study with men deprived of liberty, who were serving their sentences in a closed regime in a state prison unit in Minas Gerais (Brazil). The data collection was conducted through semi-structured interviews and processed utilizing Grounded Theory. The categories produced in this qualitative analysis using Grounded Theory were interpreted and organized based on the Patient-Centered Access Conceptual Framework. We conducted interviews with 16 men, and analysis allowed the creation and bringing closer together of the following categories related to the patient: 1) Ability to perceive 2) Ability to seek 3) Ability to reach 4) Ability to pay 5) Ability to engage, and dimensions related to the service offered: 1) Approachability 2) Acceptability 3) Availability and accommodation 4) Affordability 5) Appropriateness. For men deprived of liberty, barriers to accessing oral health result from state negligence, which promotes overcrowding and a lack of qualified professionals. This scenario creates a micropolitics of access, breaking the logic of equity.

## Introduction

Brazil has the third-largest prison population worldwide, with a vertiginous trend of continuous growth [1,2]. However, the Brazilian penitentiary system—being marked by structural and organizational disarray—has been unable to keep pace with this growth [3]. Consequently, the conditions in prison spaces are precarious and the facilities are overcrowded, precipitating unhealthy sanitary conditions; inadequate food; insufficient human resources; improvised work routines; and insufficient legal, educational, and health assistance [4].

**Data availability statement:** According to the guidelines imposed by the Brazilian National Health Council, the data collected (interviews, memos, analysis) are considered confidential because they contain sensitive information about the participants. The Free and Informed Consent Form informs these restrictions to guarantee the confidentiality and protection of the participants' information. Below is the contact information to request additional data: "You may also contact the CEP - Ethics Committee for Research with Human Beings at the Federal University of Uberlândia, located at Av. João Naves de Ávila, nº 2121, block A, room 224, Santa Mônica campus - Uberlândia/MG, 38408-100; telephone: 34-3239-4131 or by email cep@propp.ufu.br.".

**Funding:** This study was supported by CNPq - National Council for Scientific and Technological Development - INCT Saúde Oral e Odontologia - Grants n. 406840/2022-9; and by CAPES - Coordination for the Improvement of Higher Education Personnel - Grants n. 001. The founders had no role in the study design, data collection and analysis, decision to publish, or preparation of the manuscript.

**Competing interests:** The authors have declared that no competing interests exist.

In this context, although the right to health in Brazil is universal (including oral health), evidently, the incarcerated population has greater difficulty accessing healthcare [5,6]. Thus, oral diseases that typically become acute complications or generate demands for emergency services are highly prevalent [7,8].

Health care for incarcerated people demands structure and logistics inherent to security, which encompasses escorting by criminal police officers after removal from the cell, supervision, consultation, and accompanied return to the cell [8,9]. These complex logistics have aggravated the problem of access to dental care in the Brazilian context [10]. Prisoners recognize their oral health condition's precariousness through frequent painful symptoms, difficulty chewing, tooth loss, and a lack of access to oral rehabilitation, in addition to poor hygiene conditions and a lack of health promotion activities [11].

Understanding the particularities of access to health in the prison environment has enabled the development of strategies that can repair violations of the right to health and human dignity [10]. A national policy for oral health care within the Brazilian jail system was proposed. Initially, this support was the obligation of the States, which enacted initiatives within prisons without definitive national directives. In 2014, this support was incorporated into the Unified Health System (SUS) by a public policy known as the National Policy for Comprehensive Health Care for Individuals Deprived of Liberty in the Prison System. Subsequently, health initiatives and services were initiated for jailed individuals, encompassing preventive, curative, and rehabilitative aspects. The primary aim of this program is to guarantee access to healthcare within the jail environment. [9]. Therefore, this study initially assumed that the inclusion (or increase) of dental surgeons in the Brazilian prison system would have been insufficient to guarantee access to health care services. The aim of our study was to formulate a theory concerning access to oral health and the influencing factors, grounded in the perspectives of incarcerated men.

## Methodology

### Study design and ethical aspects

We conducted a qualitative study to answer the following question: "What are the meanings and interactions generated by men prisoners regarding access to oral health care?" This study was ethically assessed and approved by the Research Ethics Committee of the Federal University of Uberlândia (number 5,952,008; CAAE: 67732723.9.0000.5152).

### Sampling and study participants

Prisoners were recruited through invitations dispatched with the support of the prison health team and representatives of the prison population (who performed a support and intercommunication role) and directly to individuals with a history of experiencing oral health care inside the prison facility. The sample size was set by theoretical saturation, which could be seen in the fact that there were no more theoretically complementary categories about men's access to oral health care while they were in prison [12].

## Data collection

Data were collected in person after scheduling was agreed upon with the prison institution's management and lasted an average of 14 min and 30 s. Noteworthily, the interviews were conducted in a physical space inside the prison facility in a separate room to guarantee confidentiality and record the statements. The interviews were conducted from June 2023 until November 2023, at which point theoretical saturation was achieved [12]. This point delineated all dimensions of the theoretical framework [13,14] that were elucidated. The most challenging aspect to understand was the ability to reach dental treatment within correctional facilities. Data were collected through semi-structured, audio-recorded interviews conducted by only one interviewer (author: NSBZ), a female dental surgeon with no previous contact with the interviewees but with a history of providing care in the prison unit wherein the interviews were conducted. The researcher was previously trained by the author AMH to conduct the interviews, which used an interview guide comprising the following themes: 1) challenges faced by prisoners to access the oral health care service, 2) strategies implemented to access the service, and 3) institutional obstacles experienced by incarcerated patients in accessing health services.

Each day after conducting the interviews (five days in total), the audio-recorded content was transcribed by the author (GLS) and reviewed by another (NSBZ), removing the names and institutional references to preserve the participants' confidentiality. Interviews were numbered according to the number of the day of collection with a letter representing the order of the interviews for the day (e.g., 1A, 1B, 1C, 2A, 2B, 2C, 3A, 3B, 3C).

## Data analysis

The qualitative data produced in the semi-structured interviews were analysed using Grounded Theory. All transcribed material from the interviews was analysed in the following three coding phases: In the first phase, the transcribed material was read and coded word by word and, subsequently, line by line. At this stage, labels (codes) were created. In the second stage, the codes were compared to identify frequent and/or significant comments to understand the trajectory and access to oral health care for incarcerated patients in the penitentiary system. The codes were grouped and linked into explanatory categories of the studied phenomenon according to the dimensions of access postulated on the Patient-Centered Access Conceptual Framework. [13,14]. At this stage, gaps identified in the understanding of the phenomenon indicated questions that precipitated the need for additional interviews. Once theoretical saturation was achieved, the last stage of coding was conducted, wherein the categories organized according to the theoretical framework generated the final theorization [13,15].

## Theoretical reference

The conceptual framework for patient-centred access to health care proposed by Levesque et al. [13] was adopted to theorize the data. This framework considers access to be the result of a dynamic relationship between the characteristics of health services and users' abilities to access health care.

Regarding health services, the following five dimensions of access are understood: 1) *Approachability,* which involves the transparency, disclosure, and availability of information regarding services and access flows; 2) *Acceptability*, which pertains to cultural, economic, and social factors determining the likelihood of the patient accepting the service offered; 3) *Availability and accommodation*, which comprises geographic characteristics and mobility, waiting time, and existence of materials, and structural and human resources in sufficient volume to fulfil the patient's demands; 4) *Affordability*, which varies according to the type of service/health system and reflects the ability to cover the cost of health care; and 5) *Appropriateness*, which denotes the fit between the service and patient's needs, including the perception of technical and interpersonal quality of the service, integration and continuity of care, and the possibility of selecting providers.

Regarding the abilities and characteristics of users and families, the following aspects are considered: 1) *Ability to perceive*, which is determined by an individual's knowledge and literacy regarding one's health condition and the ability to

recognize their needs; 2) *Ability to seek,* which refers to the influence of individual, social, economic, and cultural factors on the autonomy to seek health care; 5) *Ability to reach,* which concerns personal mobility, availability of transport, and social support to reach the health care service; 4) *Ability to pay*, which encompasses the patient's ability to generate economic resources and pay for health care services; and 5) *Ability to engage,* which defines the individual's ability to make decisions and actively participate in their health care.

## Results

Sixteen men serving their sentences in a closed regime in a state prison unit in Minas Gerais State (Brazil) were interviewed. Categories were formed aligning the meanings produced by prisoners during the interviews with the dimensions of the patient-centered access to healthcare conceptual framework [13,14]. The theorization supported by theoretical reference allowed to organize the categories into two domains related to access for incarcerated men.

The first domain grouped patient abilities necessary to access health services. In our study, we named this theme "Prisoners' Abilities in Accessing Oral Health Care in the Prison System". The second assembled characteristics of health services related to access. This theme was named "Structural Barriers of the Prison System".

Table 1 presents the categories, subcategories, and illustrative statements from the theme "Prisoners' Abilities in Accessing Oral Health Care in the Prison System".

### Ability to perceive

The prisoners exhibited the ability to perceive oral health problems from the initial stages, but the search for health services seemed to begin only after painful symptoms. Pain, for them, is the result of not only their life before incarceration and the consumption of cigarettes and other drugs but also their imprisonment and restricted access to hygiene products and dental treatment.

### Ability to seek

Incarcerated individuals attribute the difficulty of access to prison overcrowding and the discrimination that they suffer. To overcome these difficulties, they promote strategies such as hunger strikes and collective protest demonstrations, including boycotting routine activities, shouting demands, and banging on bars, as well as overvaluing symptoms because they understand that only more serious cases are attended to.

### Ability to reach

Owing to the impossibility of "coming and going" freely, reaching health services requires security measures and police escort. Faced with this scenario, incarcerated individuals consider forming close ties with the criminal police and with the "free cell" person (a prisoner assigned activities within prisons) as facilitators to enable access to health services, as it prevents requests for assistance from being discarded or ignored.

### Ability to pay

The conditions for accessing hygiene products and oral health care are facilitated when the families of the incarcerated men have the financial means to pay. In Brazilian reality, when a family has sufficient financial resources, it is possible to request police escort so that the person deprived of liberty can be taken to private offices outside the prison unit. Moreover, family members can send oral hygiene products to prisoners, provided they comply with the security rules of the prison unit. Irrespective of these families' financial conditions, other payment capabilities, such as bartering medicines and other supplies, have been developed in the prison environment.

**Table 1. Categories, subcategories, and illustrative statements constituting of the theme Prisoners' Abilities in Accessing Oral Health Care in the Prison System.**

| Categories | Subcategories | Illustrative Statements |
|---|---|---|
| Ability to Perceive | Pain as the starter of the trajectory for access | "On the streets, people do not have much time to do these things because they are in the wrong life, so they do not have the time to visit the dentist, and when they get here, that is where the pain in their teeth starts to appear." (5B) |
| | | "Most people ask for a consultation because it is past time to have a tooth pulled, a filling, a restoration, or something like that." (4B) |
| Ability to Seek | Individual strategies to access the service | "Many times, because of the prisoner's desire to be treated, he invents many things. If he has a cavity, he will say that he has a hole in his tooth, is in a lot of pain, cannot sleep, and so on, but this is a lie." (1B). |
| | | "Then you had to hit the bars, had to get hungry, and had to curse the guards to get taken, but it is very difficult to get taken out. Being hungry and going two to three days without eating anything or getting care." (5E) |
| | Collective strategies to access the service | "The quickest access is to hit the bars and shout, 'Inmate feeling sick, inmate feeling sick, inmate feeling sick.' Getting the entire pavilion together means around 200 prisoners screaming simultaneously, which makes a very loud sound e but you do not always get help. (3C) |
| | | "Many times, they do not pay attention, then we [...] start shouting 'prisoner feeling sick,' then it catches their attention. Sometimes it does not work out, then we start kicking the fence and creating a ruckus and it makes a great deal of noise, so they come and pay attention." (3B) |
| Ability to Reach | Interpersonal relationships as an access mechanism | "The person who makes the note arrive here at the health unit is the 'free cell.' If he is some-what close to me and not to you, and your [case] is worse, he will definitely take my note and throw yours away (2B)." |
| | | "Many times, even the 'free cell' himself throws away the "talk to me" [written request], or the criminal police officer is still doing other things there and has no way of giving us direct attention, and sometimes, the police officer's day is also busy." (1C) |
| | Need for discipline and friendships | "Being in acquaintance with the officers makes it easier [...], a prisoner who is more isolated will have an increased difficulty of being tended to." (1B) |
| | | "If the block is undisciplined, they do not prioritize it. They do not pay attention to us at all." (3B) |
| Ability to Pay | Need for family financial support | "If a prisoner's family outside has the financial means to provide treatment, he will ask and the health sector at the unit will send him out." (3A) |
| | | "There is no condition, there is no toothbrush, there is not. If the person has a family and the family helps, they still have a situation [...] because their family helps with toothpaste, with the toothbrush." (3E) |
| | Barter to access care | "You continue being in pain, you make do with what you have, many of the times even a medicine is 'bought' in the block, from another prisoner, in exchange for cigarettes." (3C) |
| Ability to Engage | Impossibility to facilitate desired treatment | "...there are many people with toothache, there are many people who really need to have a tooth pulled, of restoration, because most people who are imprisoned in a penitentiary have their teeth destroyed, because they did not have the money to visit the dentist, there is no support inside the prison, you know, so it is only when you leave and the teeth wear out here." |

Source: Authors.

## Ability to engage

The interviewees understood that dental treatment is limited to invasive and mutilating procedures, in addition to discouraging preventive approaches. However, this curative-mutilating model of care is incompatible with prisoners' expectations. This population cannot participate in treatment decision-making and faces imposed difficulties (officialism and lack of reference services) in accessing specialized treatments.

Table 2 presents the categories, subcategories, and illustrative statements of the theme "Structural Barriers of the Prison System".

**Table 2. Categories, subcategories, and exemplifying statements integrating the theme "Structural Barriers of the Prison System".**

| Categories | Subcategories | Illustrative Statements |
|---|---|---|
| Approachability | Uncertainty regarding means of access | "Who verifies? It is right here, the office [...], but the right assortment is up to them, then they choose, for me they draw lots, it is this one, this one, and this one." (4B) |
| | | "It depends on the analysis of whoever is reading the 'talk to me' to prioritize, right? Sometimes one is more of a priority than the other, depending on the case and their need." (3B) |
| Acceptability | Punishment mediating access | "The police officer, when cross with the block, they often take it and throw it away [the 'talk to me']" (3B) |
| | Discrimination influencing access | "This is not resocializing the prisoner. This makes the prisoner more and more an animal. Because he does not have a dentist when he needs, he does not have a doctor when he needs, right away." (3B) |
| | | "It is not because we made mistakes. We are paying for what we did wrong. So I think we should not be treated like animals. Because if you stay inside a cage, you are already an animal for the state, for the state you are trash." (4B) |
| Availability and Accommodation | Overcrowding of the unit | "Because there are 1,800 prisoners here, if we are to depend on the unit here, just like I told you at the beginning there, there is no way, the person dies of toothache, they die of anger from waiting, from crying out." (4B) |
| | Lack of human resources | "Because adequate servers or professionals are lacking. Because some years ago there were none, there was a single dentist who rarely came to the unit, that was the problem." (1B) |
| | | "It is very difficult to get care here, sometimes it is not just one wing that needs care on the day, it is several and there are very few criminal police officers to attend to everyone." (3C) |
| | Resultant self-treatment | "I had to pull out the tooth myself at my own risk. As it went, I got tired. I kept loosening it, and I pulled them out by myself. It was hurting so much [...] it was a relief." (1A) |
| Affordability | Lack of state investment | "I think support from the state in this matter is lacking because here there is one dentist, just one dentist for 1,800 prisoners is very little, I think of it that way." (4B) |
| | | "If the government releases the funds for the prison to hire a dentist, I believe there will be an employee available to be able to be working, to tend to the prisoners, right?" (4E) |
| Appropriateness | Feeling of neglect with symptoms | "I think it is a great deal of medical negligence on the part of the doctors here, that prisoners arrive here feeling ill and the doctors say that they are not feeling anything, they just want to give Dipyrone [...] there is a long way to go in terms of some care to have total attention, some prisoners face a great deal of medical negligence." (4A) |
| | Lack of preventive treatment | "If, for example, if you say you want to make a cleanse, you want to have tartar removed, these people will never be able to do it." (2B) |

Source: Authors.

## Approachability

Although incarcerated individuals demonstrated clarity regarding the access mechanism (i.e., a written request, which is called "talk to me"), they did not attribute transparency in the screening and selection criteria of cases called for consultation; the interviewees conjecture different hypotheses, such as draw lots, the severity of cases, or the health professional's subjectivity.

## Acceptability

The incarcerated men believe the discriminatory treatment in relation to the prisoners originates from criminal police officers. It occurs when individuals request dental appointments, and their healthcare needs are overlooked or not communicated to the prison health team. The interviewees indicate that the absence of direct contact with healthcare professionals implies a need for mediation by the prison authorities for their care requirements. Within these procedures, criminal police officers utilize refusal of dental consultations as a punitive strategy or to decrease undesirable behavior among the incarcerated population. This obstacle is demonstrated by the neglect of written requests and the refusal of escort from the cell to the dentist's office.

### Availability and accommodation

Prisoners attributed the overcrowding of the prison unit and lack of human resources (health professionals and criminal police officers) as barriers to access, elucidating the meaning of negligence in light of this scenario. In dental care, the restricted number of vacancies and officialism for extra-mural referrals (emergencies, specialties, and private treatment) impel self-treatment measures and the acquisition of medicines without a prescription (barter).

### Affordability

Faced with the insufficiency of prison units in guaranteeing dental care, the prison population attributes the lack of financial investment to the government (or state) to overcome the lack of professionals (health professionals and criminal police necessary to provide care).

### Appropriateness

Although no objections exist regarding the technical quality of the prison facilities' dental surgeons, prisoners disagreed with the number of consultations and procedures supplied. Therefore, the delay in receiving care and lack of resoluteness mean that incarcerated people neglect their symptoms and oral health conditions.

The theorization proposed for access to dental care in the prison unit is based on the central category that the restriction of access experienced by incarcerated individuals results from negligence promoted by the state. This results in overcrowding of the prison unit and an insufficient workforce (both criminal police and health professionals). As demand is greater than supply, a micropolitics of access is established, whereby the relationship between incarcerated individuals, among themselves, and criminal police officers can intensify or facilitate access to health services. Faced with these micropolitics, strategies have been created to pressure for access (overvaluation of symptoms, hunger strikes, and collective manifestations).

## Discussion

To understand the meanings produced by prisoners regarding access to oral healthcare, the results of our study indicate that difficulties in accessing oral health in the prison system are predominantly attributed to state negligence. The form of organization and management of the prison health service and administrative bureaucracy may influence the inequality of access and does not allow the prioritization of oral health, as overcrowding and shortage of professionals resulting from negligence promote micropolitics for access.

Broadly speaking, state negligence begins even before confinement as incarcerated people have a sociodemographic and economic profile closely related to poverty, social vulnerability, and lack of access to healthcare [16]. State negligence can also be identified by the low adherence of states and municipalities to health program in prisons, demonstrating a scant effort to comply with health legislation [4,8,17]. Low adherence to national health program in the prison environment arises from the fact that adherence is not voluntary [8].

One of the themes established in this study highlights characteristics of the prison system and their negative impacts on the access to dental care of incarcerated men. This outcome relates to overcrowding, and insufficient criminal police agents and dental surgeons. Therefore, there is a repressed demand within the prison system that undermines access. Overcrowding in prison units is increasing in all Brazilian states [8]. Overcrowding of prison units is also observed in other countries, which, even after changes in the penal structure, have observed the permanence of this scenario [18]. Statistics from Uruguay suggest that the country's incarcerated population is growing continuously and that health care for people deprived of their liberty has not reached the minimum level established by international standards [19]. The same was observed in Portugal, where the growth of the prison population remained high even after penal reform measures [20]. To reduce prison overcrowding and a hyperviolent criminal ecosystem, Ecuador sought solutions through the creation of large penitentiary complexes and a review of the indiscriminate use of prison sentences; however, the results obtained were still unsatisfactory [19].

Research results from different Brazilian regions [8,21] have also linked overcrowding to limitations in access to healthcare. A study conducted in the State of Paraíba (Brazilian Northeast Region) indicated that although the number of health teams following the publication of the National Policy for Comprehensive Health Care in the Prison System (PNAISP) increased, the infrastructure of health services did not accompany the progressively expanding population demand [21,22]. The same was observed in the State of Pará (Brazilian North Region), where an increase in the absolute number of health professionals hired and prison infrastructure was still not sufficient to fulfil the labour demand of the prison units [8].

As such, the achievement of equitable access to health depends on overcoming technical and political factors that affect prison reality, configuring themselves as barriers to guaranteeing and maintaining intramural services and negatively impacting the resocialization process [11,23].

The number of health professionals [6], infrastructure, available materials, overcrowding, and precarious sanitary conditions in prison units make it infeasible to provide comprehensive and continuous dental care, even with the existence of laws that ensure access to the population deprived of freedom [11]. There is irregularity in the number of consultations conducted by prison health teams, resulting from a lack of planning and monthly production parameters according to the needs and health profile of the prison population [8,21]. The autonomy provided to states and municipalities in Brazil to implement health program in the prison environment generates disparities in the services offered, resulting in an extremely heterogeneous distribution of consultations and human resources, in addition to insufficient funding, which impairs the system's organization [11].

Structural restrictions to access oral health care found in our results have also been reported in other prisons, including those of high incomes countries. Evidence from a French study indicated that dental facilities in prisons exhibited considerable structural deficiencies, such as inadequately designed physical areas for dental practice, outdated equipment, and insufficient maintenance [24]. Moreover, challenges in inmate transportation and a lack of personnel limit dental surgeons to addressing only emergency cases [24]. Likewise, the United Kingdom, United States of America, Canada, and Australia presented deficiency of skilled practitioners, inadequate materials and equipment, and an absence of a structured care routine, which constitutes obstacles to dental treatment [25].

Research conducted in the State of Goiás (Brazilian Midwest Region) identified that incarcerated people perceive a restriction on access to oral health, considering the large number of attempts necessary to access dental care in prison units [12]. Consistently, a study conducted in Pará (Brazilian North Region) found that more than half of the prison population did not receive any type of dental care in the prison unit and that the prevalence of missing teeth was significant in all age groups assessed [4]. Difficulties in accessing oral healthcare were also reported in studies conducted in South Korea [26], Pakistan [27], Scotland [28], England [29], Finland [30], Canada [31], India [32,33], Thailand [34], Kosovo [35], and France [36], with the caveat of aggravating factors related to the condition of the incarceration itself. Noteworthily, oral hygiene supplies are insufficient, rendering the prison population dependent on religious institutions or family members to provide toothbrushes and toothpaste because the use of dental floss is almost always prohibited for safety reasons [6].

Our findings suggest that prisoners had the capacity to recognize their oral health issues. However, there is a lack of transparency and of predictability regarding the mechanisms for accessing dental care, even when pain is perceived. In other studies [35,37], incarcerated individuals recognized mechanisms of access to oral health care available in times of severe pain or obvious emergencies. Both results indicate a reactive nature of prison system, without planning of preventive or continuous care.

Even when access to oral health services is obtained, care is highly mutilating regardless of the lack of supplies to conduct rehabilitative procedures in penitentiary offices, and exacerbated medicalization to alleviate painful symptoms is common [6,27,31]. When evaluating the profile of dental service provision in Brazilian prison units, it was possible to observe a high prevalence of tooth extraction compared to other clinical treatments [10]. Despite the high prevalence of dental cavities in Bahia (Brazilian Northeast Region), a low percentage of restored teeth is observed [18].

The global panorama of oral health conditions and dental treatment needs of prisoners indicates that the burden of oral diseases is substantially greater in this group as compared to the general population, with high rates of tooth loss, dental cavities, and periodontal disease [32,34–37]. Although this study's central aim was to understand access to oral health within the prison environment, incarcerated people did not attribute their oral health conditions solely to the prison context, but also to their life before incarceration. This result aligns with what was observed [4], which indicated that crime, drug use, time, and financial resources to access health services before prison impacted oral health. Before incarceration, the use of medications and illicit drugs is common to reduce pain. However, upon entry into the prison system and deprivation of access to these subterfuges, the perception of pain becomes more evident, which explains the increased demand for dental services [11,27,35,38].

The perspective of the inmates presented in our results indicates access to dental care may be constrained by healthcare professionals' prejudices against the incarcerated population, which undermines the legitimacy of inmates' needs. This results in a micro-policy of access that reflects the relevance of interpersonal relationships to access health services (irrespective of the real need for oral health). Per Santos et. al [10], the need for criminal police officers to escort the patient throughout the entire dental treatment process creates an additional challenge, as the number of workers to perform this function amid other diverse escorting and security demands, which negatively impacts both the initial and follow-up consultations, is insufficient. In this context, there is a greater focus on tasks related to custody than on those related to healthcare, despite the existence of regulations to equalize the two functions [39]. Moreover, when inmates exhibit behavior that is considered disrespectful, criminal police officers allow themselves to employ repression, such as preventing access to health care [40].

This conduct may be shaped by ideological beliefs and ethical considerations grounded in a conservative perspective that the prison setting confines individuals who ought to forfeit all their rights, not merely their liberty. These beliefs could affect the dynamics between criminal police officers and inmates, as well as the operation of healthcare services. This scenario may have the potential to stigmatize the incarcerated population and rationalize the denial of rights, including oral health care, since prejudice diminishes the quality of health care and impedes the execution of governmental policies [41,42].

Brazil has a universal healthcare system that includes the provision of dental care to the prison population. However, the provision of health services in the prison system does not meet the demands for dental care [08]. Given the difficulty of obtain oral health care, our results showed that some inmates end up using private external services when they have ability to pay. This finding indicates the challenges of adequate financing the oral health services into the prison system which may also be observed in other countries. In South Korea, prisoners are excluded from the public health insurance system and depend only on services provided by correctional facilities. Nonetheless, the demand for care exceeds the supply [43]. In the United States, which lacks a universal healthcare system, access to dental services is limited. Individuals with a history of incarceration are less likely to keep health insurance coverage. Consequently, efforts are required to enhance health insurance coverage to mitigate the health inequities seen by this population [44].

A limitation of the study is that, for safety reasons, the interviews were conducted with open doors and in the presence of a police escort outside the room. Despite the confidentiality of the interviews, interviewees may have felt less comfortable sharing their opinions regarding the care trajectory.

## Conclusion

The meanings produced in the interviews allowed us to understand that the barriers to accessing oral health have been attributed by incarcerated men, mainly, to governmental negligence. This notion stems from the chronic scenario of overcrowding, the shortage of oral health and public safety professionals, which does not seem to be adequately addressed by the government. These problems reinforce inequalities, create a micro-politics of access, and consequently perpetuate inequity in access to oral health care for the incarcerated population.

Our results, as well as the precarious oral health conditions of the prison population, indicate the need to strengthen government policies aimed at health in the prison environment and to address overcrowding. In the realm of public policies, once the legal framework is established, it is up to the government's responsibility to promote compliance with the defined parameters and to continuously oversee their execution. The legislative definition stipulates that the quantity and working hours of dental surgeons should be appropriate with the incarcerate population, varying from 6 hours per week for up to 300 detainees to 30 hours per week for facilities housing over 1,700 inmates. Ultimately, our findings underscore the necessity to reevaluate incarceration as a punishment for minor offenses and to enhance the training of criminal police officers about human rights, particularly in relation to healthcare access.

## Acknowledgments

This work was carried out with the support of the Coordination for the Improvement of Higher Education Personnel – Brazil (CAPES).

## Author contributions

**Conceptualization:** Naessa Santos Borges Zure, Glaubieny Lourenço dos Santos, Thallys Rodrigues Félix, Jaqueline Vilela Bulgareli, Álex Moreira Herval.

**Data curation:** Naessa Santos Borges Zure, Glaubieny Lourenço dos Santos, Thallys Rodrigues Félix, Jaqueline Vilela Bulgareli, Álex Moreira Herval.

**Formal analysis:** Naessa Santos Borges Zure, Glaubieny Lourenço dos Santos, Thallys Rodrigues Félix, Jaqueline Vilela Bulgareli.

**Funding acquisition:** Álex Moreira Herval.

**Investigation:** Naessa Santos Borges Zure, Glaubieny Lourenço dos Santos, Thallys Rodrigues Félix.

**Methodology:** Naessa Santos Borges Zure, Thallys Rodrigues Félix, Jaqueline Vilela Bulgareli, Álex Moreira Herval.

**Project administration:** Jaqueline Vilela Bulgareli, Álex Moreira Herval.

**Resources:** Naessa Santos Borges Zure, Álex Moreira Herval.

**Supervision:** Álex Moreira Herval.

**Validation:** Naessa Santos Borges Zure, Glaubieny Lourenço dos Santos, Thallys Rodrigues Félix, Jaqueline Vilela Bulgareli, Álex Moreira Herval.

**Visualization:** Naessa Santos Borges Zure, Glaubieny Lourenço dos Santos, Thallys Rodrigues Félix, Jaqueline Vilela Bulgareli.

**Writing – original draft:** Naessa Santos Borges Zure, Glaubieny Lourenço dos Santos, Thallys Rodrigues Félix.

**Writing – review & editing:** Naessa Santos Borges Zure, Glaubieny Lourenço dos Santos, Thallys Rodrigues Félix, Jaqueline Vilela Bulgareli, Álex Moreira Herval.

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
