## [Decision Letter · Decision Letter 0]

13 Jan 2025

Dear Dr. Herval,

Thank you for submitting your manuscript to PLOS ONE. After careful consideration, we feel that it has merit but does not fully meet PLOS ONE’s publication criteria as it currently stands. Therefore, we invite you to submit a revised version of the manuscript that addresses the points raised during the review process.

We look forward to receiving your revised manuscript.

Kind regards,

Jessye Melgarejo do Amaral Giordani, Ph.D.

Academic Editor

PLOS ONE

Journal Requirements:

2. Please provide additional information regarding the considerations made for the prisoners included in this study. For instance, please discuss whether participants were able to opt out of the study and whether individuals who did not participate receive the same treatment offered to participants.

This study was supported by CNPq - National Council for Scientific and Technological Development - INCT Saúde Oral e Odontologia - Grants n. 406840/2022-9; and by CAPES - Coordination for the Improvement of Higher Education Personnel - Grants n. 001.  

5. Please amend either the title on the online submission form (via Edit Submission) or the title in the manuscript so that they are identical.

6. Please amend your authorship list in your manuscript file to include author Álex Moreira Herval, Naessa Santos Borges Zure, Glaubieny Lourenço dos Santos, Thallys Rodrigues Félix, and Jaqueline Vilela Bulgareli.

7. Please amend your manuscript to include your abstract after the title page.

Reviewers' comments:

Reviewer's Responses to Questions

**Comments to the Author**

1. Is the manuscript technically sound, and do the data support the conclusions?

Reviewer #1: Yes

Reviewer #2: Partly

2. Has the statistical analysis been performed appropriately and rigorously?

Reviewer #1: N/A

Reviewer #2: N/A

3. Have the authors made all data underlying the findings in their manuscript fully available?

Reviewer #1: No

Reviewer #2: Yes

4. Is the manuscript presented in an intelligible fashion and written in standard English?

Reviewer #1: Yes

Reviewer #2: Yes

Reviewer #1: The justification for the proposed study is clear and valid, as it deals with a relevant and little-discussed subject, especially as it is directly related to human rights, the violation of rights by the prison system and the violence perpetuated within state institutions.

Some adjustments to methodology and analysis are required.

Reviewer #2: This is the review of the manuscript entitled “Theorization regarding access to oral health care for Brazilian prisoners: a qualitative study.”

Thank you for the opportunity to review this work for PLOS One

Abstract.

Grounded Data Theory.

C. Change this term to Grounded Theory.

16 men were interviewed and analysis allowed the creation of the following…

C. Do not begin a sentence with numbers.

16 men were interviewed and analysis allowed the creation of the following categories related to the patient: 1) Ability to perceive 2) Ability to seek 3) Ability to reach 4) Ability to pay 5) Ability engage, and categories related to the service offered: 1) Approachability 2) Acceptability 3) Availability and accommodation 4) Affordability 5) Appropriateness

C. Here the authors refer to the units of analysis as categories, and yet, in the manuscript they are presented as codes. The authors need to make a methodological choice and decide which one to use. It is my view that, since they have used a Theoretical Framework do organize the analysis, since it comes with pre-established conceptual categories, this makes more coherent to use them as categories (Conceptual Framework for Patient-Centered Access to Health Care by Levesque et al.14). As yet for the codes, then what is called sub-codes could be held as codes, making it again coherent with the methodology used.

Brazilian government formulate a health police to improve health care in the prisons few years ago.

C. It has formulated. Please, say more about what a health police means, it is hard to follow here. Or would it be a health policy?

The aim of this study was theorizing the meanings and interactions engender by Brazilian men prisoners about access oral health care.

C. Access to oral health care.

5) Ability engage

C. This probably means Ability to engage.

The aim of this study was theorizing the meanings and interactions engender by Brazilian men prisoners about access oral health care.

C. This objective is a bit complicated. Theorizing the meanings and interactions do not necessarily contribute to a specific aim within the Health System, unless it is pointed, at least to some extent, to help explaining the main issues related to access to oral health care, which seems to be the main focus of this paper. It is my suggestion that the authors choose a more precise objective, away from this one that is too open.

The interviews took place from June/2023 to November/2023, when theoretical saturation occurred.

C. Theoretical saturation needs to be explained and supported by a relevant citation.

Data collection was performed through semi-structured audio-recorded interviews, conducted by a single interviewer (author: NSBZ)

C. conducted by only one interviewer.

Ability to pay

The conditions for accessing hygiene products and dental services are facilitated when the family of the individual incarcerate has the ability to pay for these products.

C. It is unclear if the prisoners can access dental services if they could afford it, because in the way it has been written, it sounds like they could do it. Can they access dental treatment in a private dental office?

Acceptability

It was understood that there is discriminatory treatment in relation to the prisoners originates from the discrediting of the need for dental care by the prison institution.

C. This sentence is hard to follow. Maybe the authors want to claim that ‘the discriminatory treatment in relation to the prisoners originates from’ and it has emerged in the analysis.

For them, the criminal police use preventing consultations as a disciplinary punishment method.

C. It is unclear to me if the preventing consultations or the denial to access preventing consultations is considered punishment method. It seems that not accessing preventing consultations would be the case, but it is unclear in this paragraph.

DISCUSSION

Faced to understand the meanings produced by prisoners regarding access to oral health care, the results of our results…

C. I guess the authors wanted to say the results of our study.

The values and ideals of the professionals in prison health teams also influence their actions towards the prison population, impacting in the implementation and functioning of the health service40 .

C. In which sense? This is unclear as well, as to how it could be detrimental or not. Please rewrite it, such that this becomes clear to the reader.

CONCLUSION

The meanings identified in the interviews allowed us to understand that barriers to accessing oral health have been mainly attributed to state negligence, which results in overcrowding, lack of health professionals and criminal police officers. Furthermore, the prison interactions to access health care promotes a micropolitics of access which generates inequity in oral health.

C. The state negligence results in overcrowding, lack of health professionals and criminal police officers – this may well be truth. Yet, it is a statement that perhaps goes beyond the aim proposed in this study. It might be the case that this Conclusion be more to the point of oral health care, pointing to the reasons to lack of access within the prison system. Such reasons are well described and discussed, as well. It is understandable that overcrowding is a reason, of course, but in a growing violence environment, in developing countries where social injustice is prevalent, vulnerability becomes a consquence and brings with it crimes and convictions, augmenting the incarcerated population. Thus, it is hard to deal with such a huge problem in an oral health scientific paper. Though the authors do not necessarily have to confine within the realm of oral health all the time, in the CONCLUSION it is necessary to conclude within the limits proposed by this field; hence, there is a need to produce a concise section where the main findings are brought up again, along with proposals do be implemented.

Then, the authors are also invited to produce suggestions that might be used within the system, and that could be carried out. They could also make a more intricate discussion around the legislation – especially in the sense of how many dentists should/could be employed by the system to attend such a large demand, as well.

**Do you want your identity to be public for this peer review?** For information about this choice, including consent withdrawal, please see our Privacy Policy

Reviewer #1: **Yes: ** Pires, Fabiana Schneider

Reviewer #2: No

---

## [Author Response · Author response to Decision Letter 1]

4 Mar 2025

Dear Editor,

We are pleased to have the opportunity for reviewing our manuscript. We conducted a comprehensive review of our manuscript. If, following our improvement, you decide further data and new analysis necessary, we will proceed accordingly.

ABSTRACT AND INTRODUCTION

Suggestion 1: Grounded Data Theory. C. Change this term to Grounded Theory.

Solution: We sincerely value your significant input to enhancing our manuscript. Indeed, we were employing a term that does not conform to international standards. Consequently, we will replace the term "Grounded Data Theory" with "Grounded Theory" throughout the abstract and methods sections. We made changes to lines 28 and 115.

Suggestion 2: 16 men were interviewed, and analysis allowed the creation of the following… C. Do not begin a sentence with numbers. (line 29)

Solution: We enhance our writing to prevent such blunders. The revised sentence is: We conducted interviews with 16 men, and analysis allowed the creation of the following categories related to the patient (lines 29 and 30).

Suggestion 3: 16 men were interviewed, and analysis allowed the creation of the following categories related to the patient: 1) Ability to perceive 2) Ability to seek 3) Ability to reach 4) Ability to pay 5) Ability engage, and categories related to the service offered: 1) Approachability 2) Acceptability 3) Availability and accommodation 4) Affordability 5) Appropriateness. C. Here the authors refer to the units of analysis as categories, and yet, in the manuscript they are presented as codes. The authors need to make a methodological choice and decide which one to use. It is my view that, since they have used a Theoretical Framework do organize the analysis, since it comes with pre-established conceptual categories, this makes more coherent to use them as categories (Conceptual Framework for Patient-Centered Access to Health Care by Levesque et al.14). As yet for the codes, then what is called sub-codes could be held as codes, making it again coherent with the methodology used.

Solution: We appreciate the input and agree with the suggested proposals. Consequently, at the locations where the terms "codes" and "subcodes" were utilized, they were substituted with "categories" and "subcategories." The substitutions occurred on the subsequent lines:

- Lines 156 and 157: Table 1 indicates the categories, subcategories, and illustrative statements of Prisoners’ Abilities to access oral health care.

- Lines 159 and 160: Table 1. Categories, subcategories, and illustrative statements constituting Prisoners’ Abilities in accessing oral health care in the prison system”

- Table 1, first line: “Categories and Subcategories”

- Lines 198 and 199: Table 2 presents Institutional Barriers of the Prison System, with its categories, subcategories, and illustrative statements.

- Lines 201 and 202: Table 2 Categories, subcategories, and exemplifying statements integrating the Structural Barriers of the Prison System, which limits access to oral health care in the prison system.

- Table 2, line 1: “Categories and Subcategories”

Additional Solution: We enhanced the parts of the text where the expression "Categories" was previously employed.

- Lines 153 to 155: “After the coding stages of the transcribed qualitative material, the categories emerged from the approximation of the meanings produced by prisoners in the interviews conducted using the patient-centered access to healthcare conceptual framework.

Suggestion 4: Brazilian government formulate health police to improve health care in the prisons few years ago. C. It has formulated. Please, say more about what health police means, it is hard to follow here. Or would it be a health policy?

Solution: We appreciate feedback and acknowledge that the expression "health policy" may have impeded comprehension of the content. Consequently, we have enhanced the content and will incorporate a concise section with further details regarding the specified public policy. Consequently, the subsequent paragraph has been appended.

Lines 57 to 64: A national policy for oral health care within the Brazilian jail system was proposed. Initially, this support was the obligation of the States, which enacted initiatives within prisons without definitive national directives. In 2014, this support was incorporated into the Unified Health System (SUS) by a public policy known as the National Policy for Comprehensive Health Care for Individuals Deprived of Liberty in the Prison System. Subsequently, health initiatives and services were initiated for jailed individuals, encompassing preventive, curative, and rehabilitative aspects. The primary aim of this program is to guarantee access to healthcare within the jail environment. [12]

Suggestion 5: The aim of this study was theorizing the meanings and interactions engender by Brazilian men prisoners about access oral health care.

C. Access to oral health care.

Solution: Thank you for the feedback; there was also a translation error. Nevertheless, in light of the additional ideas below, we propose a revised aim.

- Lines 23 to 25: In view of this, the aim of our study was to formulate a theory concerning access to oral health and the influencing factors, grounded in the perspectives of incarcerated men.

- Lines 66 to 68: The aim of our study was to formulate a theory concerning access to oral health and the influencing factors, grounded in the perspectives of incarcerated men.

Suggestion 6: Ability engage. C. This probably means Ability to engage.

Solution: We appreciate the feedback and concur with the recommendation. We changed the expression "Ability engage" to "Ability to engage" at all instances where the error occurred (lines: 31, 147 and 192)

Suggestion 7: The aim of this study was theorizing the meanings and interactions engender by Brazilian men prisoners about access oral health care. C. This objective is a bit complicated. Theorizing the meanings and interactions do not necessarily contribute to a specific aim within the Health System, unless it is pointed, at least to some extent, to help explaining the main issues related to access to oral health care, which seems to be the main focus of this paper. It is my suggestion that the authors choose a more precise objective, away from this one that is too open.

Solution: We value the recommendation and present a more precise objective for what is written in our article. The revised phrasing of the objective has been incorporated at end of the introduction and inside the abstract. “The aim of our study was to formulate a theory concerning access to oral health and the influencing factors, grounded in the perspectives of incarcerated men”. (Lines 23 to 25 and Lines 66 to 68)

METHODOLOGY

Suggestion 8: The interviews took place from June/2023 to November/2023, when theoretical saturation occurred. C. Theoretical saturation needs to be explained and supported by a relevant citation.

Solution: We accept with your feedback. Thus, we refine the explanation of theoretical saturation and cite two references utilized to define it.

Lines 95 to 98: The interviews were conducted from June 2023 until November 2023, at which point theoretical saturation was achieved [15]. This point delineated all dimensions of the theoretical framework [16] that were elucidated. The most challenging aspect to understand was the ability to reach dental treatment within correctional facilities.

Suggestion 9: Data collection was performed through semi-structured audio-recorded interviews, conducted by a single interviewer (author: NSBZ)

C. conducted by only one interviewer.

Solution: We are glad to accept your suggestion, and we replaced the sentence with “Data were collected through semi-structured, audio-recorded interviews conducted by only one interviewer (author: NSBZ).” (Lines 98 to 99).

RESULTS

Suggestion 10: Ability to pay. The conditions for accessing hygiene products and dental services are facilitated when the family of the individual incarcerate has the ability to pay for these products. C. It is unclear if the prisoners can access dental services if they could afford it, because in the way it has been written, it sounds like they could do it. Can they access dental treatment in a private dental office?

Solution: We understand that a more detailed explanation was lacking regarding the possibility of incarcerated men in Brazil acquiring private services. Thus, we chose to improve the wording of the topic.

Line 184 – 188: The conditions for accessing hygiene products and oral heath care are facilitated when the families of the incarcerated men have the financial means to pay. In Brazilian reality, when a family has sufficient financial resources, it is possible to request police escort so that the person deprived of liberty can be taken to private offices outside the prison unit. Moreover, family members can send oral hygiene products to prisoners, provided they comply with the security rules of the prison unit.

Suggestion 11: Acceptability. It was understood that there is discriminatory treatment in relation to the prisoners originates from the discrediting of the need for dental care by the prison institution. C. This sentence is hard to follow. Maybe the authors want to claim that ‘the discriminatory treatment in relation to the prisoners originates from’ and it has emerged in the analysis.

Solution: We appreciate the feedback, and we agree that there may have been a lack of clarity. Thus, we present a new draft for the paragraph.

- Line 212 - 219: The incarcerated men believe the discriminatory treatment in relation to the prisoners originates from criminal police officers. It occurs when individuals request dental appointments, and their healthcare needs are overlooked or not communicated to the prison health team. The interviewees indicate that the absence of direct contact with healthcare professionals implies a need for mediation by the prison authorities for their care requirements. Within these procedures, criminal police officers utilize refusal of dental consultations as a punitive strategy or to decrease undesirable behavior among the incarcerated population. This obstacle is demonstrated by the neglect of written requests and the refusal of escort from the cell to the dentist's office.

Suggestion 12: For them, the criminal police use preventing consultations as a disciplinary punishment method. C. It is unclear to me if the preventing consultations or the denial to access preventing consultations is considered punishment method. It seems that not accessing preventing consultations would be the case, but it is unclear in this paragraph.

Solution: We concur with the observation, and this issue was addressed with the prior solution. We present a new draft for the paragraph.

- Line 212 - 219: The incarcerated men believe the discriminatory treatment in relation to the prisoners originates from criminal police officers. It occurs when individuals request dental appointments, and their healthcare needs are overlooked or not communicated to the prison health team. The interviewees indicate that the absence of direct contact with healthcare professionals implies a need for mediation by the prison authorities for their care requirements. Within these procedures, criminal police officers utilize refusal of dental consultations as a punitive strategy or to decrease undesirable behavior among the incarcerated population. This obstacle is demonstrated by the neglect of written requests and the refusal of escort from the cell to the dentist's office.

DISCUSSION

Suggestion 13: Faced to understand the meanings produced by prisoners regarding access to oral health care, the results of our results… C. I guess the authors wanted to say the results of our study.

Solution: We value the feedback and concur that a drafting error occurred in this part. We now provide a revised draft of the sentence.

- Line 249 to 251: To understand the meanings produced by prisoners regarding access to oral healthcare, the results of our study indicate that difficulties in accessing oral health in the prison system are predominantly attributed to state negligence.

Suggestion 13: The values and ideals of the professionals in prison health teams also influence their actions towards the prison population, impacting in the implementation and functioning of the health service. C. In which sense? This is unclear as well, as to how it could be detrimental or not. Please rewrite it, such that this becomes clear to the reader.

Solution: We value the feedback and have revised it for clarity as below.

- Line 328-334: This conduct is shaped by ideological beliefs and ethical considerations grounded in a conservative perspective that the prison setting confines individuals who ought to forfeit all their rights, not merely their liberty. This affects the dynamics between criminal police officers and inmates, as well as the operation of healthcare services. In this setting, still significantly influenced by biases, these references exacerbate the stigmatization of the incarcerated population and rationalize the denial of rights, including oral health care. As a result, this diminishes the quality of health care and obstructs the execution of governmental policies [41].

CONCLUSION

Suggestion 14: The meanings identified in the interviews allowed us to understand that barriers to accessing oral health have been mainly attributed to state negligence, which results in overcrowding, lack of health professionals and criminal police officers. Furthermore, the prison interactions to access health care promotes a micropolitics of access which generates inequity in oral health. C. The state negligence results in overcrowding, lack of health professionals and criminal police officers – this may well be truth. Yet, it is a statement that perhaps goes beyond the aim proposed in this study. It might be the case that this Conclusion be more to the point of oral health care, pointing to the reasons to lack of access within the prison system. Such reasons are well described and discussed, as well. It is understandable that overcrowding is a reason, of course, but in a growing violence environment, in developing countries where social injustice is prevalent, vulnerability becomes a consequence and brings with it crimes and convictions, augmenting the incarcerated population. Thus, it is hard to deal with such a huge problem in an oral health scientific paper. Though the authors do not necessarily have to confine within the realm of oral health all the time, in the CONCLUSION it is necessary to conclude within the limits proposed by this field; hence, there is a need to produce a concise section where the main findings are brought up again, along with proposals do be implemented. Then, the authors are also invited to produce suggestions that might be used within the system, and that could be carried out. They could also make a more intricate discussion around the legislation – especially in the sense of how many dentists should/could be employed by the system to attend such a large demand, as well.

Solution: We value the recommendation and concur that the revision of the Conclusion was essential. We present below a rewriting conclusion.

- Lines 341-356: The meanings produced in the interviews allowed us to understand that the barriers to accessing oral health have been attributed by incarcerated men, mainly, to governmental negligence. This notion stems from the chronic scenario of overcrowding, the shortage of oral health and public safety professionals, which does not seem to be adequately addressed by the government. These problems reinforce inequalities, create a micro-politics of access, and consequently perpetuate inequity in access to oral health care for the incarcerated population.

Our results, as well as the precarious oral health conditions of the prison population, indicate the need to strengthen government policies aimed at health in the prison environment and to address overcrowding. In the realm of public policies, once the legal framework is established, it is up to the government’s responsibility to promote compliance with the defined parameters and to continuously oversee their execution. The l

---

## [Decision Letter · Decision Letter 1]

2 May 2025

Dear Dr. Herval,

Thank you for submitting your manuscript to PLOS ONE. After careful consideration, we feel that it has merit but does not fully meet PLOS ONE’s publication criteria as it currently stands. Therefore, we invite you to submit a revised version of the manuscript that addresses the points raised during the review process.

We look forward to receiving your revised manuscript.

Kind regards,

Jessye Melgarejo do Amaral Giordani, Ph.D.

Academic Editor

PLOS ONE

Reviewers' comments:

Reviewer's Responses to Questions

**Comments to the Author**

Reviewer #1: All comments have been addressed

Reviewer #2: (No Response)

2. Is the manuscript technically sound, and do the data support the conclusions?

Reviewer #1: Yes

Reviewer #2: Partly

3. Has the statistical analysis been performed appropriately and rigorously?

Reviewer #1: N/A

Reviewer #2: N/A

4. Have the authors made all data underlying the findings in their manuscript fully available?

Reviewer #1: Yes

Reviewer #2: Yes

5. Is the manuscript presented in an intelligible fashion and written in standard English?

Reviewer #1: Yes

Reviewer #2: Yes

Reviewer #1: The authors answered the highlighted points with greater theoretical depth. The topic is relevant and the research brings useful data and adequate analysis.there was an ethical commitment .

Reviewer #2: Review of ‘Theorization regarding access to oral health care for Brazilian prisoners: a qualitative study’

Thank you for the opportunity to review this work for Plos One.

ABSTRACT

The codes produced in the analysis were interpreted based on the Patient-Centered Access Conceptual Framework.

C. Categories, not codes.

Page 5

L 110. Data were collected through semi-structured, audio-recorded interviews conducted by only one interviewer (author: NSBZ), a woman dental surgeon with..

C. A female dental surgeon.

Page 6

The conceptual framework for patient-centered access to health care proposed by Levesque et al. [16] was adopted to theorize the data.

C. The authors have claimed on page 4 to have used another theoretical framework:

This study was guided by the theoretical framework of symbolic interactionism, with the understanding that the symbols and meanings of a given phenomenon emerge from an individual’s interaction with society [13,14].

C. The authors should decide which one to use, because they are not the same.

C. Now, reading the results, it becomes clear that the authors have followed the theoretical framework of patient-centered access to health care, which is ok. Then it gets confused by the fact that in the tables the authors present categories again, with their codes. It offers to the reader a comprehensive look within the raw data, which is interesting. Yet, it causes some confusion. See, if you have the categories presented in the results (of the Abstract) as: 1) Ability to perceive 2) Ability to seek 3) Ability to reach 4) Ability to pay 5) Ability to engage, and categories related to the service offered: 1) Approachability 2) Acceptability 3) Availability and accommodation 4) Affordability 5) Appropriateness.

And then, when you read the results, you find on page 11: Table 2 presents Institutional Barriers of the Prison System, with its categories, subcategories, and illustrative statements.

C. See, for the reader, this comes at the difficult task to organize in his/her mind something that is ‘Institutional Barriers of the Prison System’ without knowing what it is, really.

At least in my view, it seems that you have produced two Domains, or Themes – they can be used interchangeably [see Grounded Theory by Corbin and Strauss (Basics of qualitative research: Techniques and procedures for developing grounded theory, 3rd ed. Citation. Corbin, J., & Strauss, A. (2008).], and then the presentation should be organized under two Themes or Domains, being them (as the authors have manifested in the Abstract): … and analysis allowed the creation of the following ‘categories related to the patient’ … and then … ‘Structural Barriers of the Prison System’.

They seem to me two domains, one based on the theoretical framework used, the last one based on the classical analysis that underscores the emerging of categories in Table 2.

Therefore, such organization is demanded, so the reader can understand the Results and more, the way the theoretical framework has been used, and the way the Grounded Theory has been used to produce such results shall also be explained to avoid confusion.

Finally, I still think that the Themes, Categories and even Codes could be more thoroughly discussed in the context of the relevant literature, in the Discussion section. The authors bring interesting results with a scheme of coding and categorizing that resulted in a broad list of categories, even two Themes (in the hierarchy of Grounded Theory, one step below the creation of a Theory). Yet, the Discussion brings again a review of the literature, with no confrontation between the findings and the literature, except for the Micro-politics of Access, which is really an interesting finding and has been discussed.

So, generally the Discussion follows a sequence of discussing the categories with the relevant literature, not necessarily all categories, but the most relevant indicated by the authors. I am not criticizing the review and the information brought about within the Discussion, which is fine. What I am discussing is the fact that there is no link between the Results and this dialogue that is expected in this section.

Therefore, I hope to see a deeper contextualization of the results, the empirical results and the relevant literature in the field, beyond the general and sometimes distant topics contained in the Discussion, at this point.

**Do you want your identity to be public for this peer review?** For information about this choice, including consent withdrawal, please see our Privacy Policy

Reviewer #1: **Yes: ** fabiana schneider pires

Reviewer #2: No

---

## [Author Response · Author response to Decision Letter 2]

15 Jun 2025

Dear Editor,

We are pleased to have the opportunity to review our manuscript. We conducted a comprehensive review of our manuscript. If, following our improvement, you decide further data and new analysis are necessary, we will proceed accordingly.

ABSTRACT

Suggestion 1: The codes produced in the analysis were interpreted based on the Patient-Centered Access Conceptual Framework. C. Categories, not codes.

Solution: We appreciate the consideration given to our text. This term need to have been substituted in the prior revision. We substituted codes with categories at line 28 of the abstract.

“The categories produced in this qualitative analysis using Grounded Theory were interpreted and organized based on the Patient-Centered Access Conceptual Framework.”

DATA COLLECTION

Suggestion 2: Page 5, L 110. Data were collected through semi-structured, audio-recorded interviews conducted by only one interviewer (author NSBZ), a woman dental surgeon with. C. A female dental surgeon.

Solution: We express our gratitude for the consideration of our text. We improve our writing, and the amended sentence is “Data were collected through semi-structured, audio-recorded interviews conducted by only one interviewer (author: NSBZ), a female dental surgeon with no previous contact with the interviewees but with a history of providing care in the prison unit wherein the interviews were conducted (lines 97 and 100).

THEORETICAL REFERENCE

Suggestion 3: Page 6

The conceptual framework for patient-centered access to health care proposed by Levesque et al. [16] was adopted to theorize the data. C. The authors have claimed on page 4 to have used another theoretical framework: This study was guided by the theoretical framework of symbolic interactionism, with the understanding that the symbols and meanings of a given phenomenon emerge from an individual’s interaction with society [13, 14]. C. The authors should decide which one to use, because they are not the same.

Solution: We understand the concern regarding the mention of symbolic interactionism and the framework for patient-centered access to health care as references for the study. We assert that symbolic interactionism was identified as the foundation for formulating the research topic and the principal framework of Grounded Theory. A conceptual framework for patient-centered access to healthcare has been identified as a reference for interpreting the data. We concur with the reviewer and have chosen to eliminate symbolic interactionism from the approaches to prevent ambiguity. This judgment is founded on a thorough methodology in our research, which is not fundamentally anchored in symbolic interactionism.

Suggestion 4: C. Now, reading the results, it becomes clear that the authors have followed the theoretical framework of patient-centered access to health care, which is ok. Then it gets confused by the fact that in the tables the authors present categories again, with their codes. It offers the reader a comprehensive look within the raw data, which is interesting. Yet, it causes some confusion. See, if you have the categories presented in the results (of the Abstract as 1) Ability to perceive; 2) Ability to seek; 3) Ability to reach; 4) Ability to pay, and 5) Ability to engage, and categories related to the service offered: 1) Approachability; 2) Acceptability; 3) Availability and accommodation; 4) Affordability; 5) Appropriateness. And then, when you read the results, you find on page 11: Table 2 presents Institutional Barriers of the Prison System, with its categories, subcategories, and illustrative statements.

Solution: We begin by affirming our agreement with the theoretical framework that highlights patient-centered access to health care in our analysis. We have excluded symbolic interactionism citation from our study's. Upon reviewing the correction from the reviewer and revisiting our paper, we recognize that we failed to elucidate the integration of the framework and categories. It may be a case of input confusion in our manuscript. To address this issue, we enhance specific aspects of the process and results sections, as detailed below. We emphasized that expression categories and subcategories, rather than codes, were utilized following the most recent change.

(a) The "Data Analysis" portion of the "Methodology" undergoes enhancements to elucidate the organization of categories in accordance with the selected theoretical framework: “The codes were grouped and linked into explanatory categories of the studied phenomenon according to the dimensions of access postulated on the Patient-Centered Access Conceptual Framework. [13, 14]. At this stage, gaps identified in the understanding of the phenomenon indicated questions that precipitated the need for additional interviews. Once theoretical saturation was achieved, the last stage of coding was conducted, wherein the categories organized according to the theoretical framework generated the final theorization [13, 15].” (lines 119 and 124).

(b) The "Results" section undergoes a new introduction: “Categories were formed aligning the meanings produced by prisoners during the interviews with the dimensions of the patient-centered access to healthcare conceptual framework [13, 14]. The theorization supported by theoretical reference allowed to organize the categories into two domains related to access for incarcerated men.

The first domain grouped patient abilities necessary to access health services. In our study, we named this theme “Prisoners’ Abilities in Accessing Oral Health Care in the Prison System”. The second assembled characteristics of health services related to access. This theme was named “Structural Barriers of the Prison System”. (lines 154 and 161).

(c) Table 1 presents the categories, subcategories, and illustrative statements of the theme Prisoners’ Abilities in Accessing Oral Health Care in the Prison System. (lines 162 and 163).

(d) Table 2 presents the categories, subcategories, and illustrative statements of the theme Institutional Barriers of the Prison System. (lines 204 and 206).

Suggestion 5: The C. See, for the reader, this comes at the difficult task to organize in his/her mind something that is ‘Institutional Barriers of the Prison System’ without knowing what it is, really. At least in my view, it seems that you have produced two Domains, or Themes – they can be used interchangeably [see Grounded Theory by Corbin and Strauss (Basics of qualitative research: Techniques and procedures for developing grounded theory, 3rd ed. Citation. Corbin, J., & Strauss, A. (2008).], and then the presentation should be organized under two Themes or Domains, being them (as the authors have manifested in the Abstract): … and analysis allowed the creation of the following ‘categories related to the patient’ … and then … ‘Structural Barriers of the Prison System’. They seem to me two domains, one based on the theoretical framework used, the last one based on the classical analysis that underscores the emerging of categories in Table 2.Therefore, such organization is demanded, so the reader can understand the Results and more, the way the theoretical framework has been used, and the way the Grounded Theory has been used to produce such results shall also be explained to avoid confusion.

Solution: We thank you for the feedback. We concur that enhancing the description of our analysis and findings was essential. We hope that improvements made to answer the last suggestion are enough to clarify these aspects. We are willing to implement further enhancements, if required.

DISCUSSION

Suggestion 6: Finally, I still think that the Themes, Categories and even Codes could be more thoroughly discussed in the context of the relevant literature, in the Discussion section. The authors bring interesting results with a scheme of coding and categorizing that resulted in a broad list of categories, even two Themes (in the hierarchy of Grounded Theory, one step below the creation of a Theory). Yet, the Discussion brings again a review of the literature, with no confrontation between the findings and the literature, except for the Micro-politics of Access, which is really an interesting finding and has been discussed. So, generally the Discussion follows a sequence of discussing the categories with the relevant literature, not necessarily all categories, but the most relevant indicated by the authors. I am not criticizing the review and the information brought about within the Discussion, which is fine. What I am discussing is the fact that there is no link between the Results and this dialogue that is expected in this section.

Therefore, I hope to see a deeper contextualization of the results, the empirical results and the relevant literature in the field, beyond the general and sometimes distant topics contained in the Discussion, at this point.

Solution: We are pleased with the positive points highlighted regarding our discussion, and we have engaged in making this section better integrated with the literature. In response to reviewer notes, we have carefully revised and restructured the “Discussion” section. We have sought to systematically and critically integrate the categories presented in the results. Following, we present the additional discussion inserted.

(a) One of the themes established in this study highlights characteristics of the prison system and their negative impacts on the access to dental care of incarcerated men. This outcome relates to overcrowding, and insufficient criminal police agents and dental surgeons. Therefore, there is a repressed demand within the prison system that undermines access. (lines 267 and 270).

(b) Structural restrictions to access oral health care found in our results have also been reported in other prisons, including those of high incomes countries. Evidence from a French study indicated that dental facilities in prisons exhibited considerable structural deficiencies, such as inadequately designed physical areas for dental practice, outdated equipment, and insufficient maintenance [24]. Moreover, challenges in inmate transportation and a lack of personnel limit dental surgeons to addressing only emergency cases [24]. Likewise, the United Kingdom, United States of America, Canada, and Australia presented deficiency of skilled practitioners, inadequate materials and equipment, and an absence of a structured care routine, which constitutes obstacles to dental treatment [25]. (lines 300 and 308).

(c) Our findings suggest that prisoners had the capacity to recognize their oral health issues. However, there is a lack of transparency and predictability regarding the mechanisms for accessing dental care, even when pain is perceived. In other studies [Bergum et al. (2024) and Zajmi et al. (2018)], incarcerated individuals recognize access to oral health care available in times of severe pain or obvious emergencies. Despite the differences, our and other results indicate a reactive nature of prisonal system, without planning of preventive or continuate care. (lines 321 and 326).

(d) The findings of this study suggest that access to dental care is constrained by healthcare professionals' prejudices against the incarcerated population, which undermines the legitimacy of inmates' needs. This results in a micro-policy of access that reflects the relevance of interpersonal relationships to access health services (irrespective of the real need for oral health). (lines 344 and 347).

(e) Brazil has a universal healthcare system that includes the provision of dental care to the prison population. However, the provision of health services in the prison system does not meet the demands for dental care [08]. Given the difficulty of obtain oral health care, our results showed that some inmates end up using private external services when they have ability to pay. This finding indicates the challenges of adequate financing the oral health services into the prison system and may be observed in other countries. In South Korea, prisoners are excluded from the public health insurance system and depend only on services provided by correctional facilities. Nonetheless, the demand for care exceeds the supply [43]. In the United States, which lacks a universal healthcare system, access to dental services is limited. Individuals with a history of incarceration are less likely to keep health insurance coverage. Consequently, efforts are required to enhance health insurance coverage to mitigate the health inequities seen by this population [44]. (lines 362 and 372).

We reiterate our gratitude for the valuable contribution, which certainly strengthened the quality of the manuscript. We remain at your disposal for further improvements and explanations.

---

## [Decision Letter · Decision Letter 2]

7 Aug 2025

Dear Dr. Herval,

Thank you for submitting your manuscript to PLOS ONE. After careful consideration, we feel that it has merit but does not fully meet PLOS ONE’s publication criteria as it currently stands. Therefore, we invite you to submit a revised version of the manuscript that addresses the points raised during the review process.

We look forward to receiving your revised manuscript.

Kind regards,

Jenna Scaramanga

Staff Editor

PLOS ONE

Journal Requirements:

Additional Editor Comments:

The manuscript has been evaluated by two reviewers, and their comments are available below. While the reviewers are now satisfied with the article, there are some further minor concerns for you to address.

In Lines 355-361 several strong claims are made which do not appear to be fully supported by the study and references are not provided. To address this, please tone down all of the claims made in this paragraph, making it clear that your claims are speculative and not conclusions. For example, the first sentence could be rephrased as "This conduct may be shaped by ideological beliefs and ethical considerations grounded in a conservative perspective that the prison setting confines individuals who ought to forfeit all their rights, not merely their liberty."

In lines 379–380 you write “barriers to accessing oral health have been attributed by incarcerated men, mainly, to governmental negligence.” This is a claim well supported by your data. In other places where the text makes claims about governmental negligence, you should make clear that this is the perception of your research participants, or else cite other evidence to support the claim. For example, in lines 344–346 you write "The findings of this study suggest that access to dental care is constrained by healthcare professionals' prejudices against the incarcerated population, which undermines the legitimacy of inmates' needs." The methodology of this study cannot support this claim; the data relates to inmates’ experiences and perceptions of the care they receive. Please revise your submission to ensure your conclusions are supported by the results reported.

Could you please revise the manuscript to carefully address the concerns raised?

Reviewers' comments:

Reviewer's Responses to Questions

**Comments to the Author**

Reviewer #1: All comments have been addressed

Reviewer #2: All comments have been addressed

2. Is the manuscript technically sound, and do the data support the conclusions?

Reviewer #1: Yes

Reviewer #2: Yes

3. Has the statistical analysis been performed appropriately and rigorously?

Reviewer #1: N/A

Reviewer #2: N/A

4. Have the authors made all data underlying the findings in their manuscript fully available?

Reviewer #1: Yes

Reviewer #2: Yes

5. Is the manuscript presented in an intelligible fashion and written in standard English?

Reviewer #1: Yes

Reviewer #2: Yes

Reviewer #1: (No Response)

Reviewer #2: This is the review of the manuscript “Theorization regarding access to oral health care for Brazilian prisoners: a qualitative study”_R2.

This time I feel very satisfied with the manuscript. All my questions have been answered, and the result is a much better manuscript, in my view. I therefore recommend publication.

Thanks for the opportunity to review this work for PLOS ONE.

**Do you want your identity to be public for this peer review?** For information about this choice, including consent withdrawal, please see our Privacy Policy

Reviewer #1: **Yes: ** Fabiana Schneider Pires

Reviewer #2: **Yes: ** Renato José De Marchi

---

## [Author Response · Author response to Decision Letter 3]

11 Sep 2025

Dear Editor,

We are pleased to have a new opportunity to further improve our manuscript. We conducted additional reviews according to the suggestions below:

Suggestion 1: “In Lines 355-361 several strong claims are made which do not appear to be fully supported by the study, and references are not provided. To address this, please tone down all of the claims made in this paragraph, making it clear that your claims are speculative and not conclusions. For example, the first sentence could be rephrased as ‘This conduct may be shaped by ideological beliefs and ethical considerations grounded in a conservative perspective that the prison setting confines individuals who ought to forfeit all their rights, not merely their liberty’.”

Solution: We agree with the reviewer's suggestion, and corrections were made as suggested, with additional text modifications in lines 356, 357, 360, and 362:

“This conduct may be shaped by ideological beliefs and ethical considerations grounded in a conservative perspective that the prison setting confines individuals who ought to forfeit all their rights, not merely their liberty. These beliefs could affect the dynamics between criminal police officers and inmates, as well as the operation of healthcare services. This scenario may have the potential to stigmatize the incarcerated population and rationalize the denial of rights, including oral health care, since prejudice diminishes the quality of health care and impedes the execution of governmental policies [40, 42].

Suggestion 2: “In lines 379–380 you write “barriers to accessing oral health have been attributed by incarcerated men, mainly, to governmental negligence.’” This is a claim well supported by your data. In other places where the text makes claims about governmental negligence, you should make clear that this is the perception of your research participants, or else cite other evidence to support the claim. For example, in lines 344–346 you write, "The findings of this study suggest that access to dental care is constrained by healthcare professionals' prejudices against the incarcerated population, which undermines the legitimacy of inmates' needs." The methodology of this study cannot support this claim; the data relates to inmates’ experiences and perceptions of the care they receive. Please revise your submission to ensure your conclusions are supported by the results reported.”

Solution: We now explicitly state in line 344 that our findings stem from the inmates’ perspective, and we also reviewed the phrasing in line 346:

“The perspective of the inmates presented in our results indicates access to dental care may be constrained by healthcare professionals' prejudices against the incarcerated population, which undermines the legitimacy of inmates' needs.”

We again reiterate our gratitude for the valuable contributions and for the approval of the manuscript. We remain at your disposal for further refinements and explanations.

---

## [Editor Report · Decision Letter 3]

15 Oct 2025

Theorization regarding access to oral health care for Brazilian prisoners: a qualitative study

Short title: Access to oral health care for Brazilian prisoners

PONE-D-24-26616R3

Dear Dr. Herval,

We’re pleased to inform you that your manuscript has been judged scientifically suitable for publication and will be formally accepted for publication once it meets all outstanding technical requirements.

Kind regards,

Laura Kelly, PhD

Division Editor

PLOS One
---

## [Editor Report · Acceptance letter]

PONE-D-24-26616R3

PLOS ONE

Dear Dr. Herval,

I'm pleased to inform you that your manuscript has been deemed suitable for publication in PLOS ONE. Congratulations! Your manuscript is now being handed over to our production team.

Kind regards,

on behalf of

Dr. Laura Hannah Kelly

Staff Editor

PLOS ONE